# Patent Landscape Review on Ankle Sprain Prevention Method: Technology Updates

**Natrisha Francis** *⊙, **Aziz Ong, Hazwani Suhaimi** *⊙ **and Pg Emeroylariffion Abas** ⊙

Faculty of Integrated Technologies, Universiti Brunei Darussalam, Jalan Tungku Link,
Bandar Seri Begawan BE1410, Brunei
* Correspondence: natrisha.francis@gmail.com (N.F.); hazwani.suhaimi@ubd.edu.bn (H.S.)

**Abstract:** Ankle sprains are among the most prevalent musculoskeletal system injuries. Prevention of ankle sprains is important, given the frequency of occurrence of an ankle sprain, the risk of reinjury, and its long-term effects. A systematic patent review using the World Intellectual Property Organization (WIPO)'s PATENTSCOPE database has been performed to study the current development of ankle sprain prevention methods. Using the PRISMA statement as a basis, a total of 426 patent documents have been selected for review, consisting of 150 granted patents and 276 patent applications. In the past decade, China and the United States of America (43% and 29% of the patent applications, respectively) have shown great interest in developing ankle sprain prevention methods. Approximately 49% (or 74) of the patents from the 150 granted are associated with braces, 46% (or 68) are related to orthosis, 3% (or 5) are related to tape, and the remaining 2% (or 3) are associated with other types of ankle sprain prevention methods. This patent review shows that inventors are leaning towards braces and orthosis as the main prevention methods for ankle sprains, with greater interest in orthosis in recent years. Additionally, patents on smart wearable devices suggest initial commercial interest in the development of smart wearable devices as an ankle sprain prevention method.

**Keywords:** ankle sprain prevention; ankle brace; ankle orthosis; patent review





## 1. Introduction

Ankle sprains are the most common musculoskeletal injuries, especially in the athletic world [1]. Ankle injury is one the most common injuries sustained during sports (34.3%), and ankle sprain is the most common ankle injury, making up 76.7% of all the ankle injury studies considered in the review [1]. Fong et al. [1] found that ankle sprain incidents were predominantly high in court games and team sports, such as rugby, soccer, volleyball, handball, and basketball. Doherty et al. [2] estimated that indoor/court sports incidence rates were as high as 7 ankle sprains per 1000 exposure. Out of the 43 sports analysed [1], 40 sports reported ankle sprain as one of the common injuries, with it being the most prevalent in 33 of the 40 reported sports.

There are three types of ankle sprain injuries: inversion, eversion, and high ankle sprain. The most frequent ankle sprain injury mechanism is supination, which involves the inversion of the plantarflexed foot. Explosive inversion or supination, which often happens at the subtalar joint, can cause a tear of the lateral ligament complex, especially the anterior talofibular ligament (ATFL) [3]. Inversion injuries constitute about 25% of all injuries of the musculoskeletal system [4], with about 50% of these injuries sports-related; since running, jumping, and turning are common in most sports, high stresses are often put on the lower limb joints, including the ankle. Improper treatment of inversion injuries of the ankle can result in persistent ankle problems. Athletes with a history of ankle sprains are often associated with chronic ankle problems, such as chronic pain and muscular weaknesses [5]. Even though research has produced advancements in the prevention, diagnosis, and management of ankle injuries, epidemiology studies have revealed that ankle inversion ligamentous sprain remains a dominant sports injury [2].

Prevention of ankle sprains is important, given the frequency of occurrence of an ankle sprain, the risk of reinjury, and its long-term effects. There have been several reviews made on preventive methods for ankle sprain [6–9] in the literature; however, none has reviewed the subject by looking into patent filings. With the wide availability of methods that have been proposed for the prevention of ankle sprains, this patent landscape aims to provide an overview of the current wearable preventive methods for ankle sprains by conducting a systematic patent landscape review via a patent database. This looks into the patents filed on the technology at different patent offices worldwide. A patent review gives a clear picture of the technological advancements of a given field and aids in the development and implementation of a long-term research and development plan that considers different aspects of the technology [10,11]. Although not all patents result in active commercialization, analysis of published patents over the years can provide information for inferring interest in this area [12]. The patent landscape review on ankle sprain prevention devices has been performed by adopting the PRISMA statement [13] as a basis. Relevant keywords related to the topic were selected for performing the search. The resulting patent documents from the keyword search were filtered to remove patents of the same patent family and irrelevant patents, with the remaining patent documents analysed and results compared with the academic literature. Patents have to undergo stringent examinations by specialised examiners in the specific technology to ensure the novelty, inventive steps, and industrial applicability of the patented invention before it can be granted. As such, the patent landscape provides information on interesting developments in ankle prevention technology, especially from the commercial and development perspective.

## 2. Background

### 2.1. Ankle Sprains

Ankle sprains refer to damage due to the stretching or tearing of the ankle's ligaments and can be classified depending on the extent of damage to the ligament. Signs and symptoms of ankle sprains can be helpful for the examination and assessment of the extent of the injury. The following sub-subsections describe the classification and types of ankle sprains. The long-term effects of ankle sprains, including chronic ankle instability and recurrent injury, are also discussed in the later sub-subsection.

2.1.1. Classifications of Degrees of Ankle Sprain

Ankle sprains may be graded as grade I, II, or III, depending on the severity (Table 1). Determining the injury severity at the time of injury is difficult due to pain and swelling; thus, clinical grading is subjective and made based on the amount of pain, swelling, and bruising [14]. Clinically, simple sprains (Grade I) do not require anything more than asymptomatic treatment, whereas severe sprains involving mechanical instability (Grade II and III) may require additional treatment [15].

**Table 1.** Classification of ankle sprain injuries (Adapted from ref. [16]).

| Grade | Description | Signs and Symptoms |
|---|---|---|
| I | Partial tear of a ligament | • Mild tenderness and swelling<br>• Slight or no functional loss (i.e., still able to bear weight and ambulate with minimal pain)<br>• No mechanical instability |
| II | Incomplete tear of a ligament, with moderate functional impairment | • Moderate pain and swelling<br>• Mild to moderate ecchymosis<br>• Tenderness over involved structures<br>• Some loss of motion and function (i.e., pain with weight-bearing and ambulation)<br>• Mild to moderate instability |
| III | Complete tear and loss of integrity of a ligament | • Severe swelling<br>• Severe ecchymosis<br>• Loss of function and motion (i.e., unable to bear weight and ambulate)<br>• Mechanical instability |

2.1.2. Types of Ankle Sprain

There are three types of ankle sprains—lateral, medial, and high—with lateral ankle sprains constituting the most common type of ankle sprain [2].

High ankle sprains (also known as Syndesmotic sprains) involve injury to the ligaments of the tibiofibular syndesmosis. These ligaments are commonly injured during skiing, ice hockey, and soccer [17]. High ankle sprains occur during hyperflexion and extreme external rotation of the foot [18]. As the anterior and posterior tibiofibular ligaments are very stable, a sprain of these ligaments is often associated with bone fractures [19] and can be relatively severe. Due to this, high ankle sprains are found to require the longest amount of recovery out of the three types of ankle sprains [20]. In the general population, high ankle sprains account for only 1–17% of ankle sprains [21]; however, among athletes, they contribute to approximately 30% of ankle sprains [22].

Medial ankle sprains usually involve injury of the deltoid ligament. Nearly all medial ankle sprains are accompanied by other ligamentous injuries and fibular fractures, and isolated injuries to the deltoid ligaments only are quite rare [23]. A combination of dorsiflexion and eversion, accompanied by high velocities of both movements, results in medial ankle sprains. It has also been observed that the ankle is externally rotated, with a minor plantarflexion and a moderate level of eversion, during initial contact with the ground [24]. Gulbrandsen et al. [25] suggested that medial ligament sprains account for approximately 9.77% of ankle injuries in football. The highest rate of deltoid ligament sprains among collegiate athletes were in women's gymnastics, men's and women's soccer, and men's football; however, the rate for women's gymnastics was imprecise [26].

Lateral ankle sprains are the most common type of ankle sprain of the three types of ankle sprains, involving about 25% of all injuries of the musculoskeletal system, with about 50% of these injuries sports-related [4]. Delahunt et al. [27] defined a lateral ankle sprain (also known as an inversion ankle sprain) as 'an acute traumatic injury to the lateral ligament complex of the ankle joint as a result of excessive inversion of the rear foot or a combined plantar flexion and adduction of the foot'. A total of 73% of lateral ankle sprains are due to a rupture or tear of the ATFL [28], which has been reported to be the weakest and most frequently injured ligament of the ankle [29]. In 1997, Garrick [30] was the first to suggest the typical mechanism of acute lateral ankle sprain as inversion, plantarflexion, and internal rotation. In an attempt to develop a comprehensive understanding of the mechanisms of lateral ankle sprains in football, Andersen et al. [31] reviewed videotape recordings of 26 ankle sprains in Norwegian and Icelandic elite football from the 1999–2000 season. They found that the two most frequent injury mechanisms were: (1) player-to-player contact, with impact by an opponent on the medial aspect of the leg just before or at foot strike, resulting

in a laterally directed force causing the player to land with the ankle in a vulnerable, inverted position, or (2) forced plantar flexion, where the injured player hit the opponent's foot when attempting to shoot or clear the ball [31]. Numerous studies on lateral ankle sprains have generally presented a rapid increase in inversion and internal rotation, with or without plantarflexion [32], as a probable cause of lateral ankle sprain.

### 2.1.3. Effects of Ankle Sprain

During ankle sprains, damage to the ligaments of the foot initiates changes to the joint's biomechanics and modifies neural control of the foot [33]. In addition, neuromuscular control and functional performance may be altered after an ankle sprain injury, which affects the stabilisers needed to correctly stress-shield the joint [34]. This creates a negative feedback loop, increasing the probability of a reoccurrence of ankle sprains. It has been reported that individuals with a history of acute ankle sprain have an approximately 3.5 times greater risk of sustaining another ankle sprain than those with no such history [34]. Athletes with reoccurring ankle sprains also reported mild pain and statistically significant deficits in foot proprioception and static and dynamic balance [34].

### 2.2. *Academic Literature on Ankle Sprain Prevention Methods*

Prevention of ankle sprain is important, given the frequency of ankle sprain occurrence, the risk of reinjury, and its long-term effects. The table below summarizes a previous systematic review of ankle sprain prevention. Most of the prevention methods suggested in Table 2 fall under one of two categories: external support (including braces and tape) and training programs. Both have been proven to reduce the prevalence of ankle sprains and functional ankle instability. Both methods are also used in rehabilitation programmes after an ankle sprain injury.

**Table 2.** List of Systematic Review of Academic Literature on Ankle Sprain Prevention.

| Name | Year | Title | Prevention Method |
|---|---|---|---|
| Kaminski, Needle and Delahunt [6] | 2019 | Prevention of Lateral Ankle Sprains | • External Prophylactic Support <br> ○ Taping and Bracing <br> ○ Alternative Taping and Foot Orthosis <br> • Prophylactic Prevention Programs <br> • Secondary Prevention: <br> ○ Treatment of Initial Ankle Injury |
| Evans and Clough [7] | 2012 | Prevention of ankle sprain: A systematic Review | • Bracing <br> • Taping <br> • Orthotics |
| Verhagen and Bay [9] | 2010 | Optimizing ankle sprain prevention, a critical review and practical appraisal of the literature | • Shoe Type and/or Shoe Design <br> • Taping <br> • Bracing <br> • Neuromuscular Training |
| Thacker, Stroup, Branche, et al. [8] | 1999 | The Prevention of Ankle Sprains in Sports | • Shoes and Taping <br> • Bracing <br> • Training |

External ankle supports are commonly used among athletes as a preventive measure against ankle sprains. The use of external support has been shown to have the most consistent effect on reducing ankle sprain risk associated with physical activity [35]. The two main types of external support are braces and tape, and they were first introduced in the 1880s and 1890s, respectively [6]. Primarily, there are three categories of ankle braces: soft, semi-rigid, and rigid. Semi-rigid braces include Aircast® braces, hinge braces, and lace-up braces, and they are more commonly used in athletic settings. However, some studies [36,37] have considered a lace-up brace as a soft brace. On the other hand, rigid braces include one-piece braces, stirrup braces, and unilateral shell braces [38].

Taping techniques and selections depend on the clinician's familiarity with the technique and the athlete's preference. Standard taping techniques include stir-ups, figure-8, and heel-lock techniques. Generally, the cost of taping is 3 to 25 times higher than the cost

of bracing, based on the type of activity, the number of tape applications, and the number of participating individuals [39]. Applied tape can loosen during physical activity, especially when using the white cloth tape that is commonly used in athletic settings; however, there are other kinds of tape, including self-adherent tape, that clinicians can use instead, which are more resistant to movements. There are also alternative taping methods that are available. One of the more well-known alternatives is Kinesio taping, which is different from standard taping, as it involves the precise placement of 3–4 strips of the Kinesio tape across the ankle joints in line with ankle stabilising muscle. Studies have found that the use of Kinesio taping is ineffective in improving postural control and balance to prevent ankle sprain [40,41]; however, it may be an option to help improve ankle function [42].

Other than ankle brace and ankle taping, some consider ankle orthosis as a third type of external ankle support that can be used to prevent ankle sprains. However, compared to ankle brace and ankle taping, there are not as many studies concluding the effectiveness of ankle orthosis in preventing ankle sprains. Evan and Clough only identified three studies on ankle orthosis in preventing ankle sprains, but no data on injury prevention were obtained [7]. The vagueness of the definition of ankle orthosis may also contribute to the uncertainty of the prophylactic effects of ankle orthosis. Elattar et al. [43] have defined ankle orthosis as 'an externally applied apparatus that can be inserted in a shoe to help support or improve the function of the foot and ankle' by 'offloading high-pressure areas, minimising shear forces, cushioning sites of tenderness, correcting flexible deformities, or provide foot control and support'. Evan and Clough [7] reviewed different types of orthotics, including cushion column shoes, orthotic shoes, and ankle-stabilising orthosis, with ankle orthotics hypothesised to prevent ankle sprains due to their ability to reduce inversion angles. Some researchers, such as Evan and Clough [7] and Thacker et al. [44], have included orthosis as one of the methods to prevent ankle sprains. However, some researchers, such as Kaminski et al. [6], have excluded orthosis as a prevention method in their review. Similarly, Verhagen and Bay [9] also excluded orthosis in their study, but instead included shoe type and shoe modifications as a method to prevent ankle sprains.

Training programs are another intervention that may be used to prevent ankle sprains. They are more cost-effective than external ankle supports, but more labour-intensive. Studies have reported that training programs can reduce the risk of ankle sprains by 30% to 45% [6,45,46]. Training programs decrease the incidence of ankle sprains by improving the joints' ability to adapt to a changing environment in preparation for a reaction to potential injury to the ankle [46]. These training programs aim to improve proprioception, joint stability, and neuromuscular control. Exercises of these programs can incorporate stretching, balancing, and power and agility techniques, and these preventive exercise programs are usually incorporated into an athlete's warm-up and exercise routine [6].

However, the prophylactic effects of external ankle support and exercise/training programs are naturally reduced if the athlete does not fully comply with the preventive interventions. Compliance is a term used to indicate whether the athlete correctly follows and adheres to the prescribed intervention [47]. It is hypothesised that the effects of preventive proprioceptive incentives might be larger if the level of compliance is greater [46].

In their review paper, Kaminski et al. [6] also discussed three emerging ankle sprain prevention techniques: (a) advancement of ankle brace design with lighter, semi-rigid materials that improve comfort, whilst at the same time, providing multiplane stability; (b) implementation of exercise programs via the Internet and smartphone apps; and (c) incorporation of dual-task and cognitive loading in intervention techniques.

## 3. Methodology

A systematic search of patent documents via the World Intellectual Property Organization (WIPO)'s PATENTSCOPE (https://patentscope.wipo.int/, accessed on 5 September 2020) has been conducted from the earliest archives to 5 September 2020. The PATENTSCOPE database provides access to International Patent Cooperation Treaty (PCT) applications in full-text format, as well as access to patent documents of participating national and regional

patent offices. It has over 97 million patent documents, including 4.1 million published international patent applications (PCT). The keyword strings, shown in Table 3, have been used in the database's advanced search for ankle sprain prevention methods, with the search limited to contents on the first page of the patent documents only, which includes the title and abstract.

**Table 3.** Keywords used for different categories of ankle sprain prevention methods.

| Keyword Terms | | | | |
|---|---|---|---|---|
| 'Ankle' | AND | 'Sprain* OR Injur* OR Ligament OR Strain' | AND | 'Tap* OR Brac* OR Ortho* OR Cast OR Device OR Reduc* OR Splint OR Stabilizer OR Strap* OR Support OR Shoe* OR System OR Preven* OR Protec* OR Proprioception OR Wrap' |

The '*' is a wildcard operator used to search for terms in PATENTSCOPE with 0 or more characters replaced either in the middle of the term or at the end of the term.

The PATENTSCOPE online database allows users to download a csv file of information of the patents document from the search. For this patent review, the list of patent documents was downloaded on 5 September 2020 for further filtering and analysis of the patent documents. Consequently, patents added to the database after 5 September 2020 were not included in this patent review.

Using the PRISMA statement [13] as a basis, the patent review process shown in Figure 1 has been followed. The identification stage of the search retrieved a total of 1350 patent documents. Patent documents that were from the same patent family were removed. A patent family is a collection of patent applications covering the same or similar technical content and sharing the same priority date. Patent documents were manually checked, and parent patents were kept for further analysis. 215 patent documents that were from the same patent families were manually removed from the results. The patents' titles, abstracts, and claims were extracted and thoroughly assessed to investigate the relevance of the patents on ankle sprain prevention methods before further analysis. From the remaining 1135 patent documents, 709 patent documents were further excluded, as they were deemed irrelevant upon manual examinations. These include patents for ankle rehabilitation devices; inventions that prevent other ankle injuries, excluding ankle sprains; and inventions that indirectly prevent ankle sprains (for example, the design of a brake pedal for a vehicle). The remaining 426 patent documents were classified into granted patents and patent applications. In total, 276 documents are patent applications, and the patents that have been granted came to a total of 150 documents.

The earliest identified patent document was from the year 1902. The results were limited to patent documents available in the WIPO's PATENTSCOPE database, which includes non-English patent documents, as WIPO provides English translations of the patent documents from other languages.

For patent documents qualified for this patent review, years, country, applicants, and inventors have been analysed. Granted patents are categorised into four types: braces, tape, orthosis, and others. Braces and tape are the two most common external prophylactic ankle support used to prevent an ankle sprain. For this review, an orthosis is defined as a shoe modification or wearable addition to a shoe that supports and improves the foot or ankle function, whilst others refers to patents that claim to prevent ankle sprains, but are not categorised under braces, tape, or orthosis.

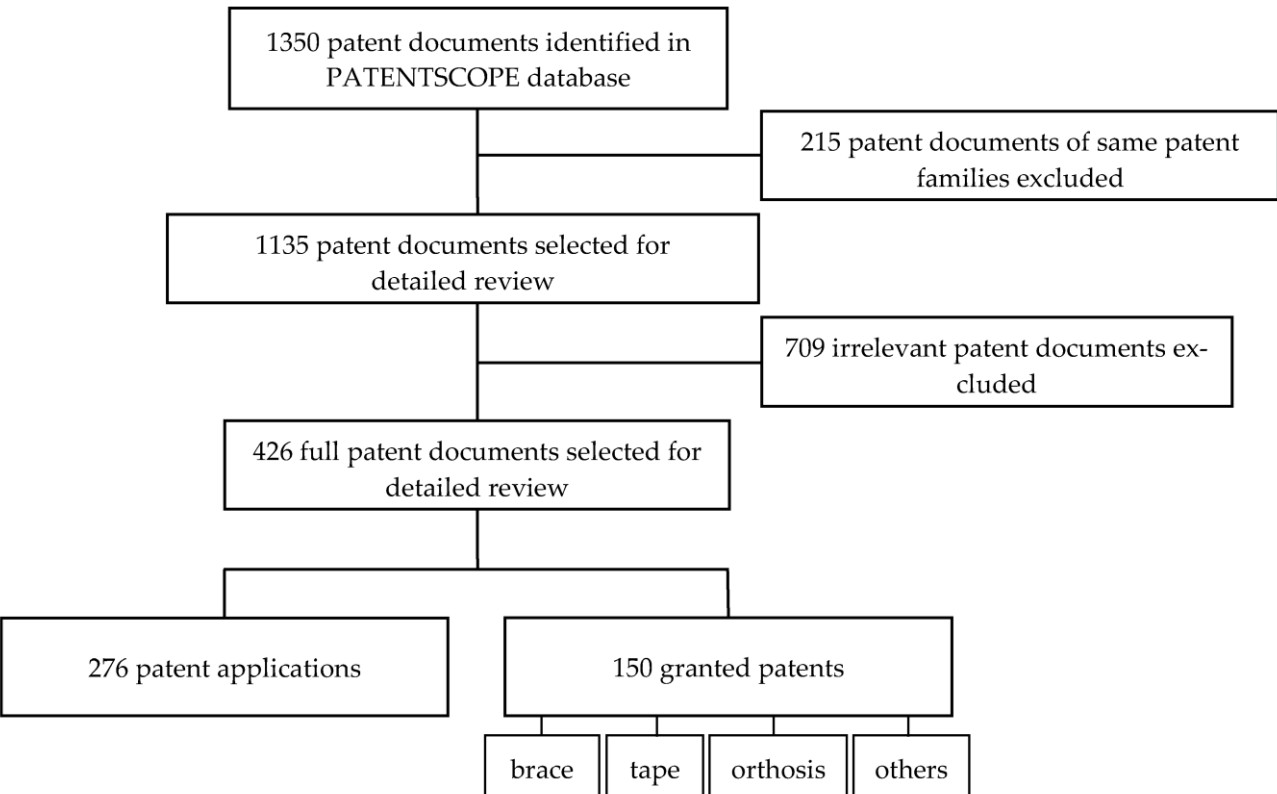

**Figure 1.** Flowchart to review patent results based on PRISMA Statement.

It is important to note that the classification of patents is very subjective. Even in academic reviews, the definitions of orthotics are somehow vague and may vary from one author to another. Elattar et al. [43] have defined ankle orthosis as 'an externally applied apparatus that can be inserted in a shoe to help support or improve the function of the foot and ankle' by 'offloading high-pressure areas, minimising shear forces, cushioning sites of tenderness, correcting flexible deformities, or provide foot control and support'. Evan and Clough [7] reviewed different types of orthotics, including cushion column shoes, orthotic shoes, and ankle-stabilising orthosis, with ankle orthotics hypothesised to prevent ankle sprains due to their ability to reduce inversion angles. Some researchers, such as Evan and Clough [7] and Thacker et al. [44], have included orthosis as one of the methods to prevent an ankle sprain. However, some researchers, such as Kaminski et al. [6], have excluded orthosis as a prevention method in their review. Similarly, Verhagen and Bay [9] also have excluded orthosis in their paper; instead, they have included shoe type and shoe modifications as a method to prevent an ankle sprain. This may be because there are not enough studies to conclude the effectiveness of ankle orthosis in preventing ankle sprains. In addition, WIPO's PATENTSCOPE database uses a translating tool specifically designed to translate patent document texts from other languages to the English language. However, despite the advanced features of the tool, specific nuances of the text may still be mistranslated, which are especially true for patents from China. For instance, the granted patent for orthosis invented by X. F. Weng [48] titled 'Temporary moulding socks for outdoor sprain and method thereof' was classified as an orthosis in this review; however, it may be considered a brace by others.

## 4. Results and Discussion

### 4.1. Patent Applications and Granted Patents on Ankle Sprain Prevention Methods

A total of 426 patent documents have been selected for a detailed review, where 276 (65%) are patent applications, and 150 (35%) are granted patents. Patent applications represent the majority of patents relevant to ankle sprain prevention methods. A patent

is granted by a national patent office or a regional office that carries out the task for several countries based on the novelty, inventive steps, and industrial applicability of the invention. Generally, patenting an invention can be expensive, as it depends on many factors, including the nature of the invention and its complexity. Furthermore, once a patent is granted, maintenance or renewal fees must be paid regularly to maintain the patent's validity. Patent applications that have not been granted may be because the owners have decided to stop pursuing the patent application due to different reasons, including insufficient funds and decisions on the ability to commercialise a product, or the patents may simply be awaiting iterations of the examination process, as the examination and re-examination process may sometimes take a long time to be completed

The number of patent documents according to their publication year can be seen in Figure 2. The blue bars in the graph indicate the number of granted patents published in that particular year and granted in later years, while the orange bars indicate the number of patent applications published in that particular year, but that have yet to be granted. From Figure 2, there is clearly an exponential increase in the number of patent applications on ankle sprain prevention methods over the years, proving an increasing interest in developing a system for the purpose. Despite the apparent increase in the number of publications, the number of granted patents has remained low throughout the years, with the exception of the earlier years considered. This may be due to the patent application awaiting the examination process or the patent application being abandoned altogether. It is noted that only patent applications from the earliest archives up to the year 2018 have been analysed, as patent documents generally take time to be granted after publication.

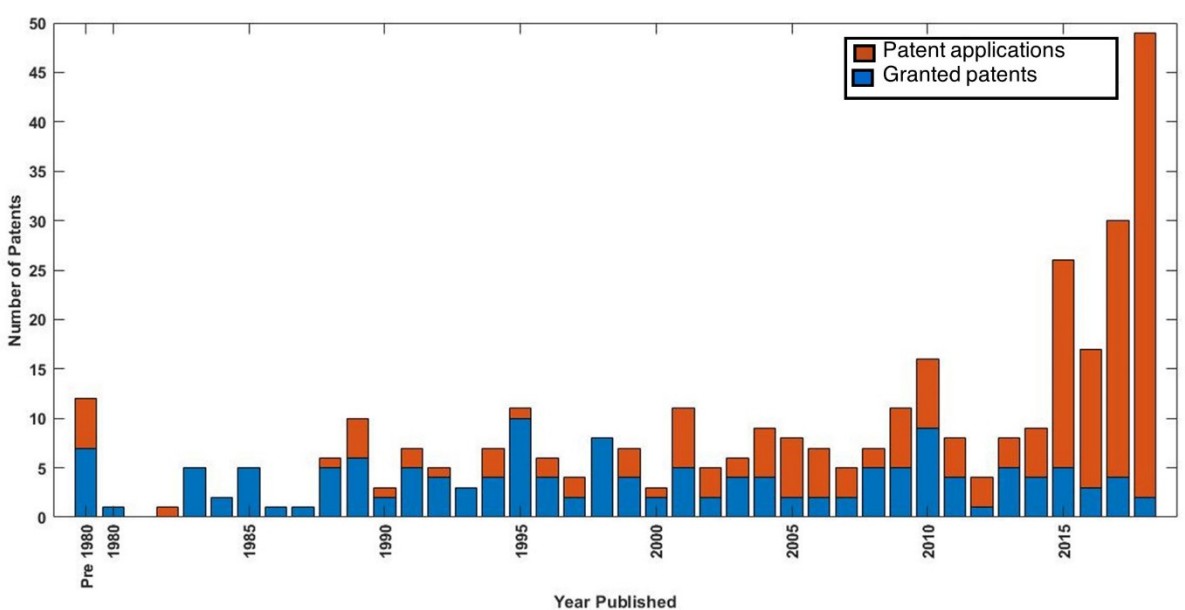

**Figure 2.** Number of patent documents according to their publication year.

A total of 150 patents have been granted over the years. The number of yearly granted patents from pre-1980 to the year 2020 showed no definite trend, with at most 9 patents granted per year, which were in 1995 and 2012.

### 4.2. Patents on Ankle Sprain Prevention Methods, Based on Countries

Figure 3a shows the different countries that have published patents on ankle sprain prevention methods. China published the greatest number of patents, accounting for 43% of the overall published patents. This is followed by the United States of America, Japan, and then Korea, with 29%, 9%, and 6%, respectively. Others_1 refers to countries that published only a single patent, which includes South Africa, Australia, India, Mexico, Netherlands, Philippines, Portugal, Serbia, and Russia.

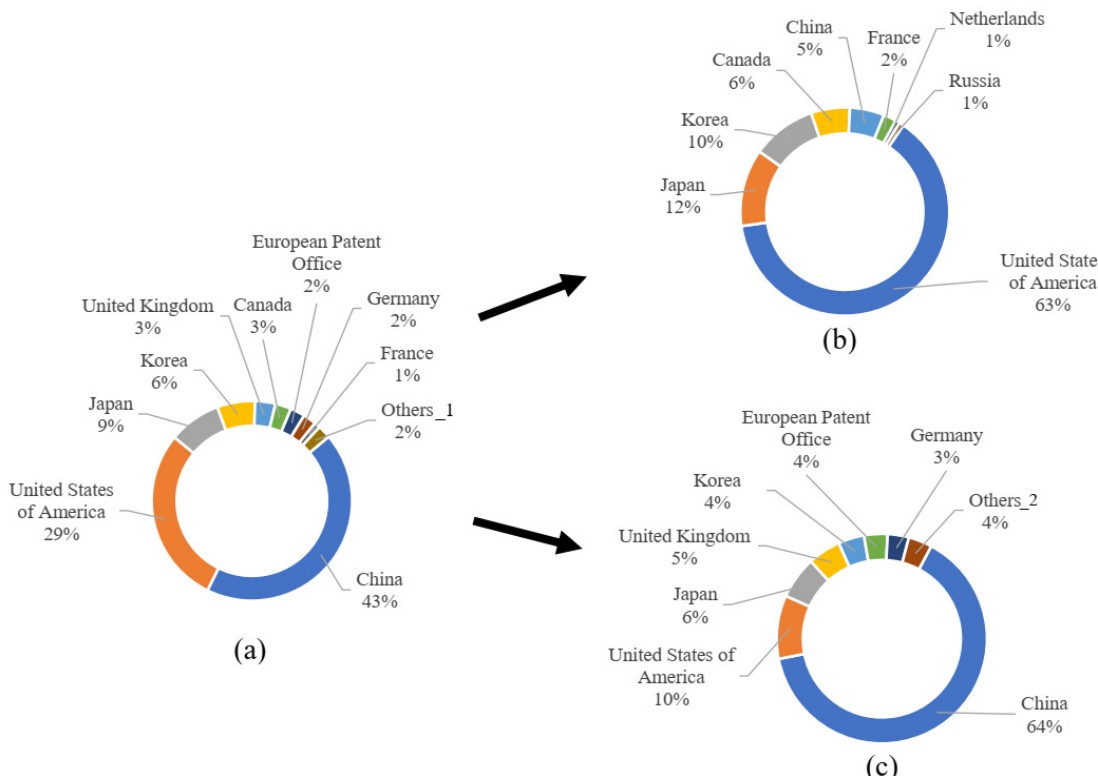

**Figure 3.** (**a**) Published patents based on countries, (**b**) Granted patents based on countries, and (**c**) Published but not yet granted patents based on countries, on ankle sprain prevention methods.

Although China has published the greatest number of patents, the United States of America has the greatest number of granted patents, as shown in Figure 3b, with a total of 95 granted patents, which represents 63% of the total granted patents, followed by Japan (12%), Korea (10%), Canada (6%), China (6%), France (2%), Netherlands (1%), and Russia (1%). The United States of America is currently the leading country, with the highest number of granted patents in the development of ankle sprain prevention methods. Shown in Figure 3c is a pie chart of the countries that have published patents that are yet to be granted. Others_2 here refers to a combination of different countries, such as Canada, which has published only two patents, and Australia, France, India, Mexico, Philippines, Portugal, Serbia, and South Africa, which have published only one patent. The majority (about 64%) of the published, but not granted patents on ankle sprain prevention methods are from China, followed by the United States of America (10%) and Japan (6%).

The number of published and granted patents by the two most active countries, China and the United States of America, by publication year are shown in Figure 4a,b, respectively. Although China has published the greatest number of patents (according to Figure 3a), it is observed in Figure 4a that China only started publishing patents on ankle sprain prevention methods in 2009, hence its relatively smaller percentage of granted patents (as seen in Figure 3b), with an exponential increase in publications in the later years. This contrasts with the United States of America, which has been publishing and granting patents on ankle sprain prevention methods as far back as before 1980, as can be seen in Figure 4b. A feasible reason for China's recent interest in patents may be due to the national strategy plans implemented by the Chinese government, including the *Outline of the National Medium- and Long-Term Science and Technology Development Plan (2006–2020)* [49], which proposes the goal of reaching the top five in the world in terms of the annual number of patents granted to nationals by 2020, and the *National Patent Development Strategy (2011–2020)* [50], which sets a quantitative goal of quadrupling the number of patent applications by 2020. China's recent interest in patenting ankle sprain prevention methods could also explain the smaller

percentage of patents granted (as shown in Figure 3b), as it normally takes time for patents to be granted. However, a comparison of the data shown in Figure 4a,b shows that the percentage of published patents that were later granted was significantly lower for China than for the United States of America. Luan and Zhang [51] have suggested that the China National Intellectual Property Administration, previously known as the State Intellectual Property Office of China, is slower at patent examination due to the overwhelming growth of patent filings in China.

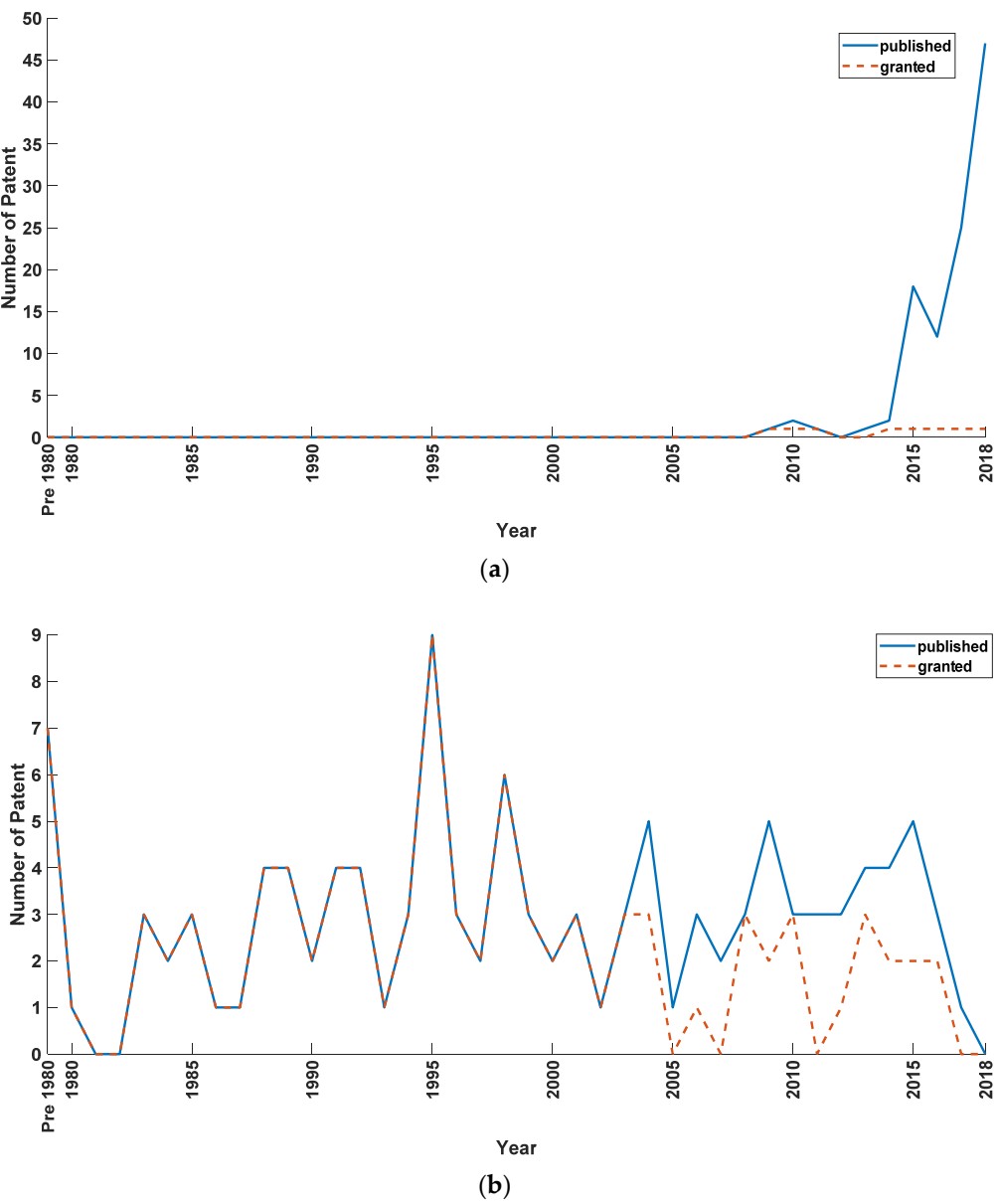

**Figure 4.** (**a**) Number of published and granted patents by China, and (**b**) Number of published and granted patents by the United States of America, according to publication year.

It has also been recorded that the highest number of patents granted were in the years 1995 and 2012. Further analysis has shown that the United States of America was responsible for granting 100% of the patents in the year 1995, while granted patents in the year 2012 were from the United States of America, with approximately 34% of the total patents, followed by Korea (33%), Japan (22%), and then China (11%). The shift in countries publishing and granting patents on ankle sprain prevention methods over the years implies a corresponding shift in the competitive market for ankle sprain prevention

products, which was previously dominated by the United States of America, but is now becoming more global.

### 4.3. Inventors and Applicants of Patents on Ankle Sprain Prevention Methods

A patent may be invented by an individual or a group of people. There is a total of 236 names found that have been listed as inventors for the granted patents. Figure 5 shows the names of inventors of two or more granted patents, with the remaining inventors classified as others. It can be seen that Nelson Ronald E. has the greatest number of granted patents on ankle sprain prevention methods and, hence, is a notable forerunner in inventing ankle sprain prevention methods, with a total of eight granted patents. All granted patents by Nelson Ronald E. are on ankle braces. His first patent was published in 1988, while his most recent patent was published in 2006, in which he has been recorded as the inventor, with Mueller Sports Medicine Inc. as the applicant of the patent. Of his eight granted patents, six patents have been filed in the United States of America and two patents in Canada.

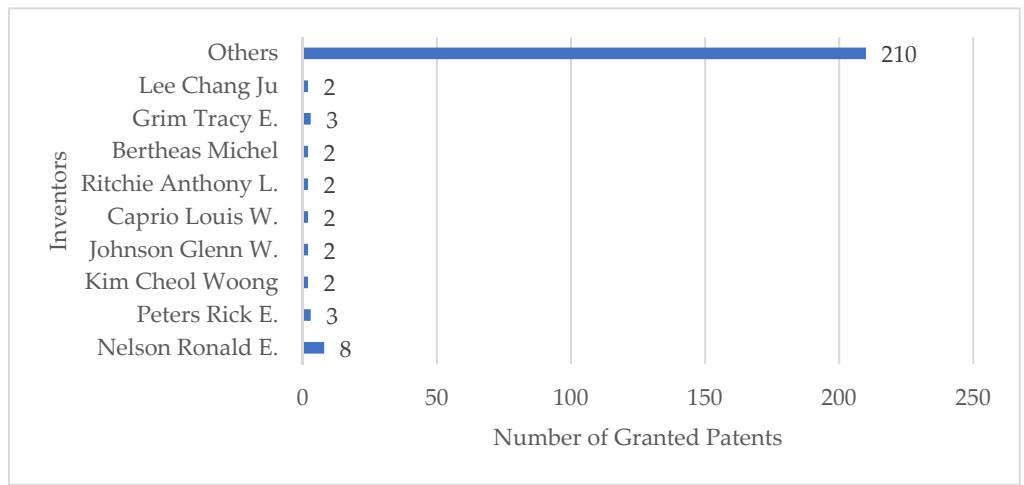

**Figure 5.** Inventors of granted patents on ankle sprain prevention methods.

Patent applications may come from an individual company or university. Categories of applicants of granted patents are shown in Figure 6a. Around 60% of the applicants are individuals, 36% are from companies, and 4% are from universities, indicating more interest among individuals in developing ankle sprain prevention methods than companies and universities. Generally, the higher numbers of patent applicants from individuals indicate early-stage technology. Individual inventors normally intend to form start-ups to develop their inventions further to a commercialisable stage, before bigger companies start taking an interest in the inventions and granted patents, either through assignment or licensing of patents or takeover of the original start-ups.

Figure 6b shows individual applicants involved in two or more granted patents, with the rest of the individual applicants classified as Others_1. The individual applicant with the greatest number of granted patents on an ankle sprain prevention method is Nelson Ronald E., with a total of seven patents. He is both the sole inventor and an applicant of 7 out of 8 granted patents associated with his name, published between the years 1988 and 1993. Company applicants involved in two or more granted patents are shown in Figure 6c, with Others_2 referring to company applicants that have only one granted patent. Active Ankle Systems Inc. has the greatest number of granted patents on ankle sprain prevention methods amongst the list of companies. Originally from the United States of America, Active Ankle System Inc. has been developing high-functioning foot and ankle products since 1989. The company was acquired in 2008 by Cramer Products, a leading company in the American sports medicine industry.

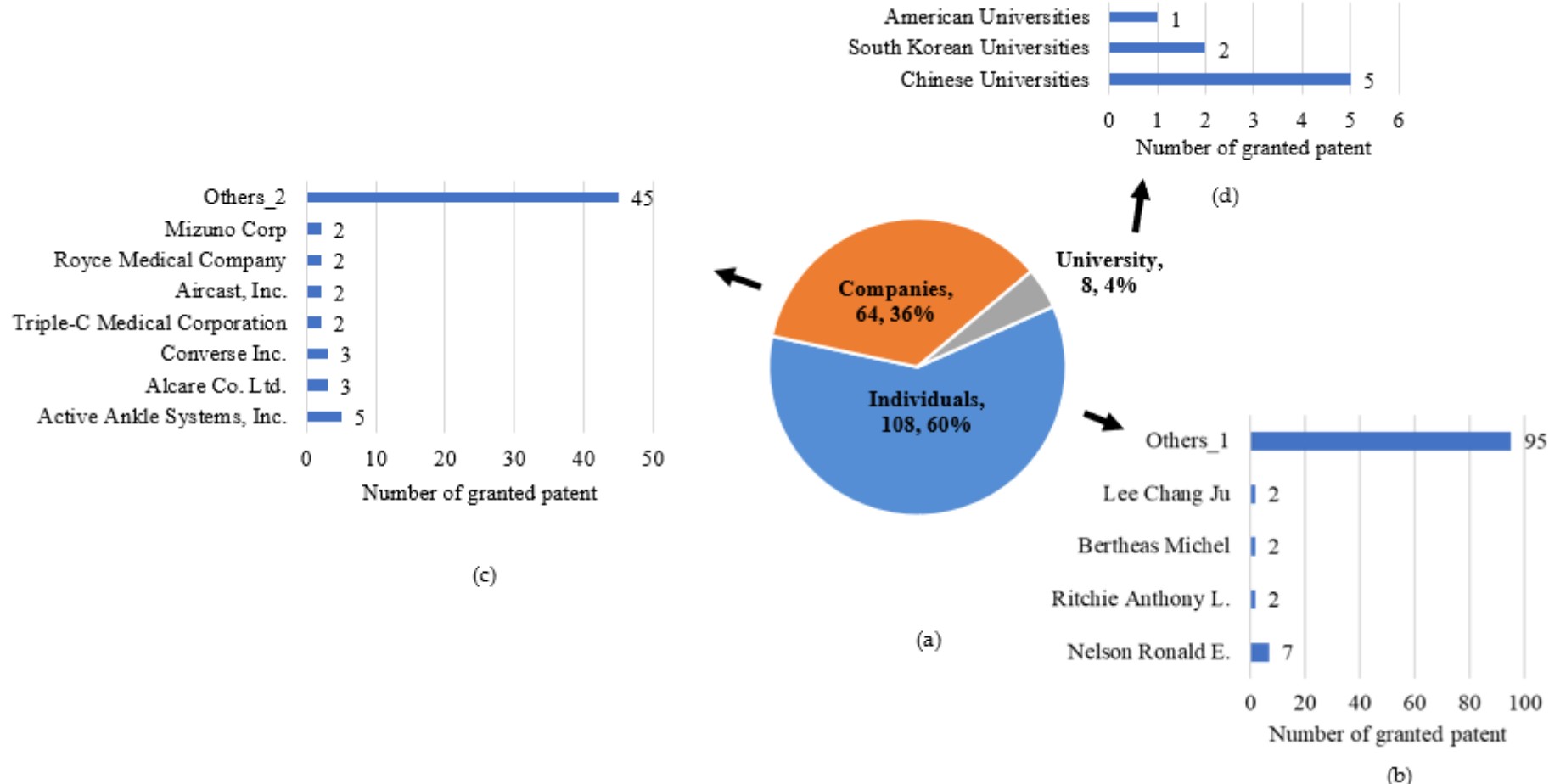

**Figure 6.** (**a**) Categories of applicants of granted patents, (**b**) Individual applicants of granted patents, (**c**) Company applicants of granted patents, and (**d**) Countries of university applicants of granted patents, on ankle sprain prevention methods.

Figure 6d shows the countries of university applicants of granted patents of ankle sprain prevention inventions, including foundations and trustees associated with a university. Out of the eight universities listed as applicants for granted patents in ankle sprain prevention methods, five are universities from China, suggesting more interest and funding for universities in China to pursue research and development of ankle sprain prevention methods. Interest amongst Chinese universities in patenting ankle sprain prevention methods may be due to the government innovation policy in the form of a "Chinese Bayh–Dole Act" and research evaluation system, which encourages researchers to patent their inventions [52]. However, past data have suggested that Chinese universities are putting greater importance on submitting patent applications, rather than producing high-quality patents [53], hence the lower proportion of granted patents.

### 4.4. Types of Ankle Sprain Prevention Methods

Finally, the patents on ankle sprain prevention methods are categorised into braces, orthosis, tape, and others. According to Figure 7, approximately 51%, or 76, of the 150 granted patents are associated with braces; 44%, or 66, are related to orthosis; 3%, or 5, are related to tape; and the remaining 2%, or 3, are associated with other types of ankle sprain prevention methods. Inventions that have been classified as orthosis and braces are more common than taping methods. This is unexpected, as orthosis is not as well-researched as bracing or taping. Evans and Clough [7] have identified three scientific articles on orthosis; however, no data on injury prevention were given. Although the effectiveness of orthosis in the prevention of ankle sprains has not been adequately documented, inventions classified as orthosis are more protected than ankle tapes. It has been found that two of the three patents classified under others are on modifications to sporting equipment, while the other one involves myoelectrical stimulation. No prevention training method for the prevention of ankle sprains has been patented.

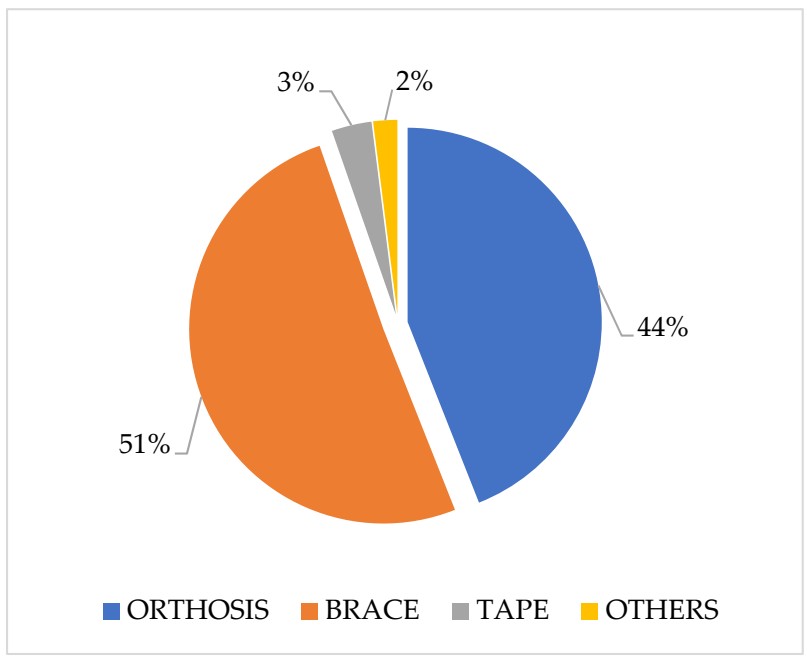

**Figure 7.** Type of ankle sprain prevention method.

Of the 5 patents on tape, 1 was published in 1973, 3 were published in 1989, and 1 was published in 2003. For patents on others, 1 was published in 1972, 1 was published in 2001, and 1 was published in 2010. The number of granted patents, according to the publication years, for braces and orthosis are shown in Figure 8a,b, respectively. The number of granted patents on braces peaked in 1995, with 8 granted patents (as shown in Figure 8a), while the number of granted patents on orthosis peaked later in 2010, with 5 granted patents (as

shown in Figure 8b), indicating a shift in focus by inventors towards ankle orthosis as an ankle prevention method, away from braces and tape.

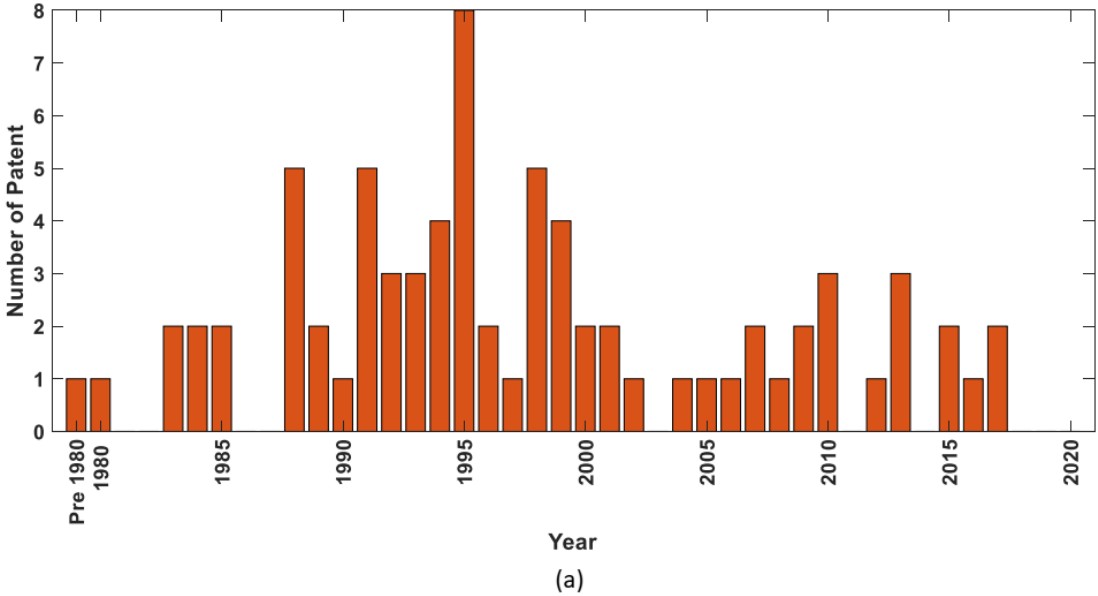

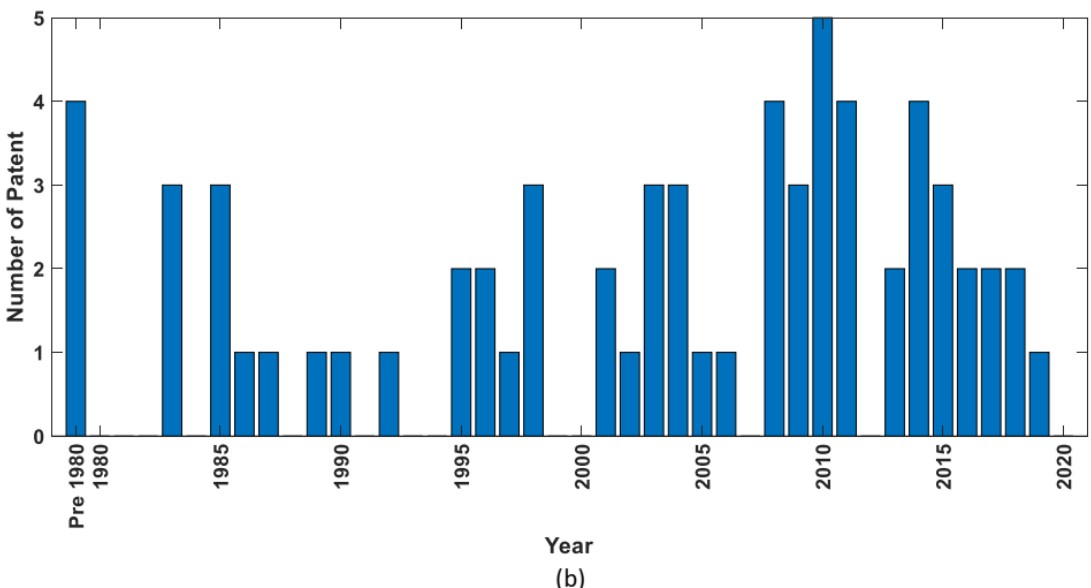

**Figure 8.** (**a**) Number of granted patents on braces, and (**b**) Number of granted patents on orthosis, according to publication year.

The patents on orthosis can be further categorised into two groups: general use and sports use. In total, 37.3% (25 out of 67) of the granted patents on orthosis are explicitly for sports, with some patents designed for multiple sports activities. The number of orthosis patents meant for a specific sports activity is shown in Table 4. There are a total of 14 different sports specified in the granted patents, with the greatest number of patents on orthosis specifically for general athletics. There are a considerable number of patents on orthosis for American football and baseball.

**Table 4.** The number of granted orthosis patents associated with the type of sport.

| Type of Sport Activity | Number of Granted Patents on Orthosis |
| --- | --- |
| General Athletics | 6 |
| American Football | 4 |
| Baseball | 4 |
| Basketball | 2 |
| Bicycling | 1 |
| Football | 3 |
| Golf | 1 |
| High Jump | 1 |
| Ice skating | 3 |
| Inline skating | 1 |
| Mountain climbing | 2 |
| Running | 1 |
| Taekwondo | 1 |
| Tennis | 3 |

## 5. Technology Updates

With increased interest in ankle sprain prevention methods, the progress in research and development, new findings, and innovative technologies has moved rapidly over the years. This is evident from the volume of patent publications and grants analysed in this review. In this section, a few interesting patents are compared and discussed in some detail to understand more about the technological advancement in ankle sprain prevention methods.

### 5.1. Selected Patents on Tape

Only 5 patents, or 3% of the overall granted patents, are of the tape type, with the earliest and latest patents on tape granted in 1973 and 2003, respectively. The earliest patent on tape was invented (or proposed) by E. Wise in 1973 [54]. It describes a simple inelastic, but flexible strap, which is secured under tension to an annular band surrounding the lower portion of an athlete's ankle and bearing to the upper portions of the foot and heel. On the other hand, the latest patent on tape was an 'Ankle Wrapping System' invented (or proposed) by A. L. Ritchie in 2003, as shown in Figure 9. The tape [55] is described as a wrapping system that includes at least one resilient section, an adhesive portion, and scored areas. It is designed to fill the recessed areas of the joint to ensure that the tape will be secured while, at the same time, providing adequate pressure over the injured joint.

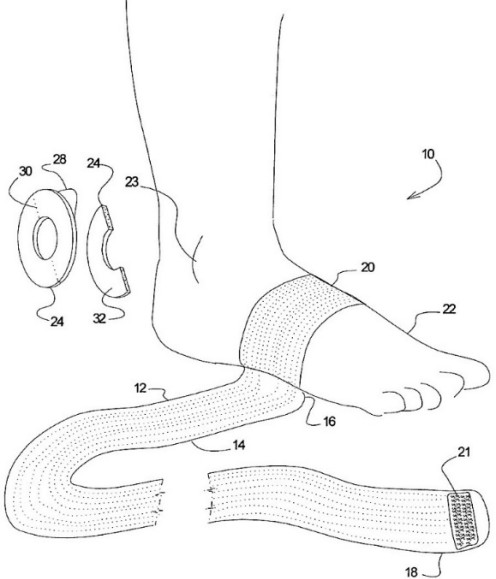

**Figure 9.** Ankle Wrapping System [54].

### 5.2. Selected Patents on Braces

Patents of the type brace represent the highest number of granted patents on ankle sprain prevention methods. Braces were a very popular technology for intellectual property protection, with a peak number of patents on braces granted in the year 1995. However, over the later years, there was a shift in interest in development towards orthosis.

The earliest granted patent document on ankle sprain prevention devices was a brace, invented by Robert C. Mann and published in 1972 [56]. It describes a stiff, open-fronted heel boot to brace the ankle against sprains, without constricting the ankle when walking. Its brace has an elastic strap section spanning the wearer's instep and a non-elastic ankle-wrapping section to hold the brace comfortably on the wearer's foot. Figure 10 depicts the drawings of the patent.

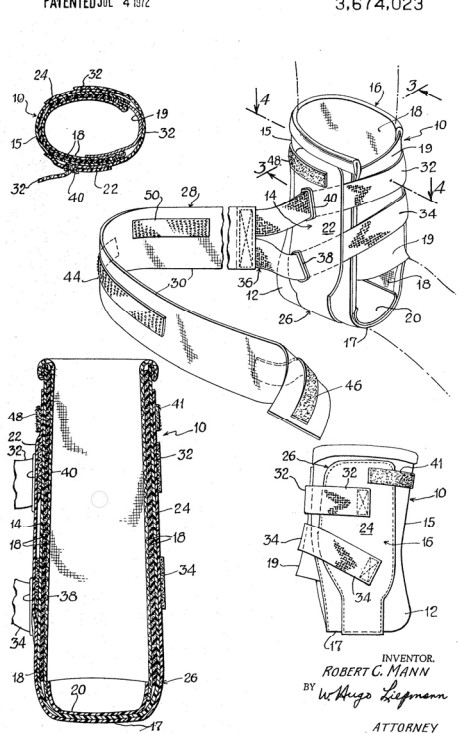

**Figure 10.** Earliest Patent on Brace: Ankle support providing high bracing strength [56].

A brace design proposed by A. D. Davis [57] in the year 1994 describes an adjustable pressure cast for orthopaedic injuries (Figure 11a). The cast has a nonrigid external sleeve, with an attached lining, inflatable longitudinal air chambers to support the lower leg, as well as inflatable air chambers to support the ankle on both sides. This brace is to be used to prevent further injury after sustaining an ankle sprain. The main advantage of the invention is its ability to control the amount of support given to different parts of the limb while the patient wears the cast. Another design of a soft brace was invented by L. Caprio in 1999 [58]. The therapeutic elastic support is made of a multidirectional fabric that is stretched around a joint to provide constant compression on the body part when used, as shown in Figure 11b. It also has a patch of a flexible laminate of closed-cell foam consisting of a heat-retaining layer. This design allows the application of heat treatment to the specific part of the injured joint.

A different design for a brace is given in Figure 11c, which is designed to prevent further injury while the wearer is healing from previous injuries. This invention was proposed by T. E. Grim [59]. It is a walker that consists of a sole with an open heel portion at the rear, a pair of rigid struts extending from the sole to just below the knee, soft supports that extend around the lower leg and foot, fasteners, and a removable inner ankle bootie whose interior is partially covered with inflatable support bladders. The soft supports are

sufficiently flexible and, hence, can accommodate swelling and dressings on the injured leg. Additionally, the device has been designed to be lightweight and easily removable, and it can be reduced to become a bootie, allowing the wearer to remain using the device comfortably in bed while sleeping.

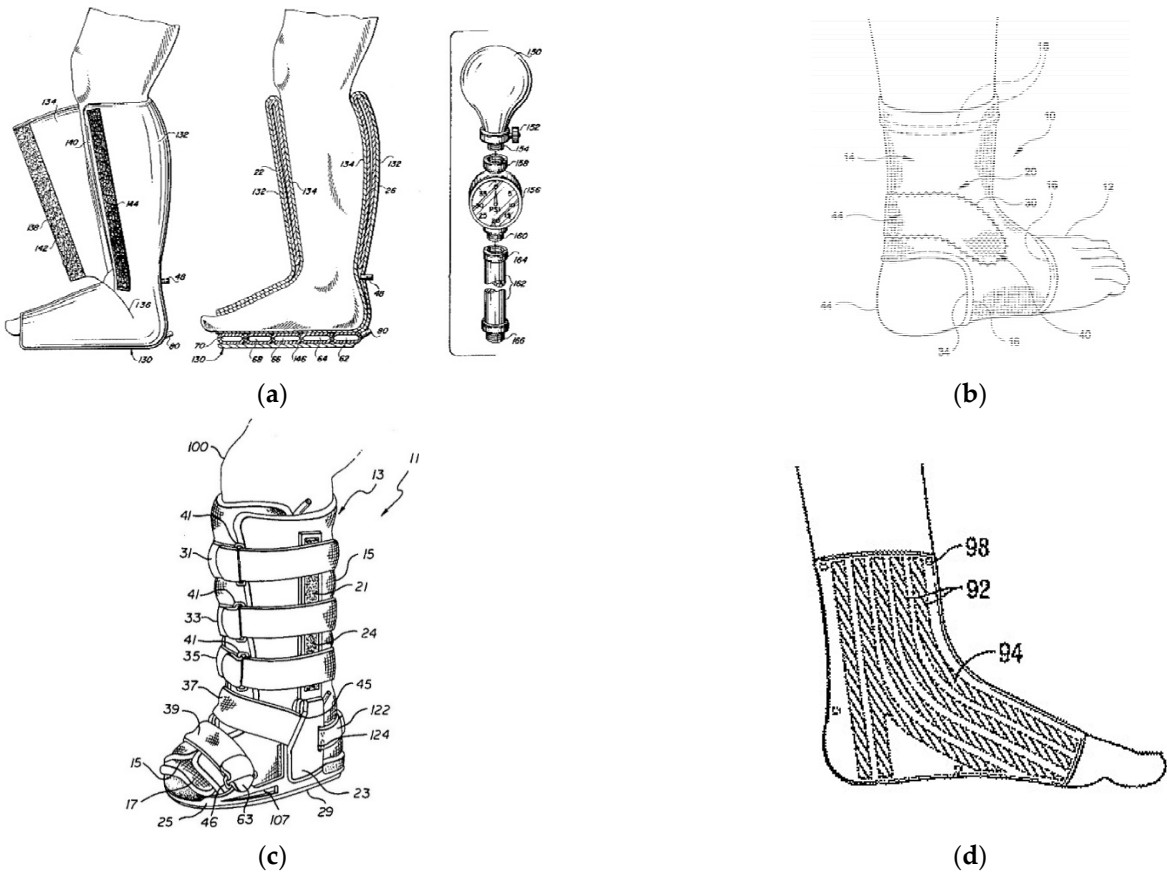

**Figure 11.** Patents of Brace: (**a**) Adjustable Pressure Cast for orthopaedic injuries [57], (**b**) Therapeutic Elastic Body Support [58], (**c**) Walker with Open Heel [59], and (**d**) Transformable Orthopaedic Brace for injury prevention [60].

The latest patent on braces was invented by J. F. J. McGuckin in 2015 [60]. The brace is a body with a series of liquid-filled regions, as shown in Figure 11d, consisting of at least one sensor positioned on the body to detect the external force experienced by the body. When the external force exceeds a predetermined value, a chemical reaction occurs, causing the liquid in the liquid-filled regions to harden, thereby limiting the joint movement of the wearer. The use of liquid-filled regions ensures the flexibility of the brace, while at the same time, the ability to automatically activate the liquid on an as-needed basis in response to force detection serves as ankle sprain protection for the wearer.

*5.3. Selected Patents on Orthosis*

Patents on orthosis are the second most granted patent under ankle sprain prevention methods, making up 45% of the granted patents analysed in this review. Despite only being the second most granted patent for preventing ankle sprains, current trends in inventions have been shifting towards orthosis.

One such orthosis for general use was invented by W. Thais and W. Kauth in 1985 [61]. It is described as a shoe with an ankle protector consisting of a support panel along the lateral side that is kept secure with adjustable straps, as shown in Figure 12a. The strap adjustment prevents inversion or internal rotation of the ankle and, thus, prevents ankle sprains. Published in 2004, J. D. Harry et al. invented an orthosis for general use [62]. The

wearable system is capable of applying neurological stimulation to the foot and the ankle via surface electrodes or vibrational actuators in or on the wearable platform, as shown in Figure 12b. A non-deterministic random signal, generated by a bias signal generator that is coupled to a controller, is used to drive the actuators and electrodes. Neurological stimulation enhances the neurosensory stimuli of the foot and ankle and, thus, enhances human balance and gait, which in turn reduces the likelihood of ankle sprains. The latest patent on a general orthosis was invented by X. F. Weng [48]. It is a sock comprising a sock body and a cloth piece, with the sock body consisting of a bottom plate and an anti-bending block fixed to the wearer with belts and fasteners, as shown in Figure 12c. The bottom plate and the anti-bending block are made of high polymer graphene material, making the device lightweight, but robust. The lower part of the anti-bending block and the foot-fitting surface of the bottom plate are covered with latex pads. Additionally, the sock also contains a storage bag to keep a cooling agent, which can either be dry ice or alcohol liquid, to allow icing of an injured ankle.

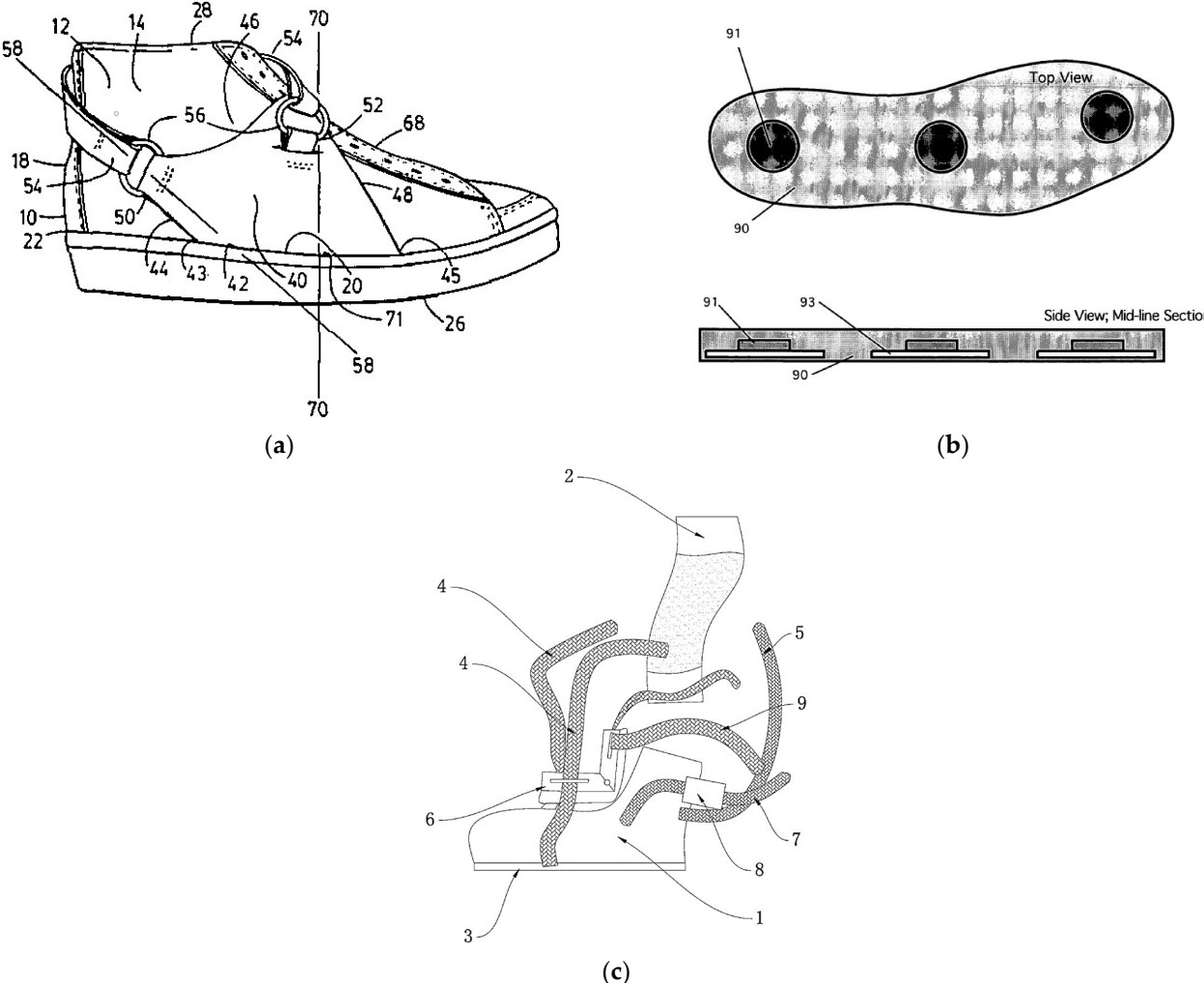

**Figure 12.** Patents of general use Orthosis: (**a**) Shoe with Ankle Protector [61], (**b**) Method and Apparatus for Improving Human Balance and Gait and Preventing Foot Injury [62], and (**c**) Temporary Moulding Sock for outdoor sprain and using method thereof [48].

In 1998, G. Haslauer invented a modified sports shoe for general athletics [63]. The sports shoe has reinforcements, with a stirrup element that encloses the foot and ankle joints, and a sleeve element that partially surrounds the calf above the ankle joint, as seen in Figure 13a. The articulation of the sleeve and stirrup on the lateral side of the foot is

arranged at an offset towards the front. This allows sufficient mobility while, at the same time, protecting the ankle from injury. The orthosis can also be used with orthopaedic socks or fabric supports for the ankle. An athletic shoe used for basketball and football was invented by E. O. Giese and R. J. Brown in 1983 [64]. The athletic shoe has an injection-moulded intermediate portion bonded with a fabric upper portion, a rubber outsole portion, an elastic band around the area of the subtalar ankle joint, an elastic collar at the ankle opening, and a T-shaped section at the top of the elastic collar, as shown in Figure 13b. The injection mould portion conforms to the metatarsal and heel area of the wearer's foot, which gives lateral support to the instep area of the wearer's foot. Additionally, the T-shape section encloses the calf just above the malleoli, thus preventing further inversion of the foot. Another sports shoe was invented (or proposed) by R. M. Parracho and K. J. Crowley [65], as shown in Figure 13c, to prevent ankle injury during running. It has a heel lift portion made of a more rigid material than the heel centre area. Its outsole is extended at the sides of the heel portion and interconnects the outsole and a heel counter member to stabilise the heel counter. The design increases the heel area's lateral stability, which consequently increases ankle stability and reduces the possibility of ankle fatigue due to misalignment of the ankle joint. Collectively, these reduce the risk of ankle sprains.

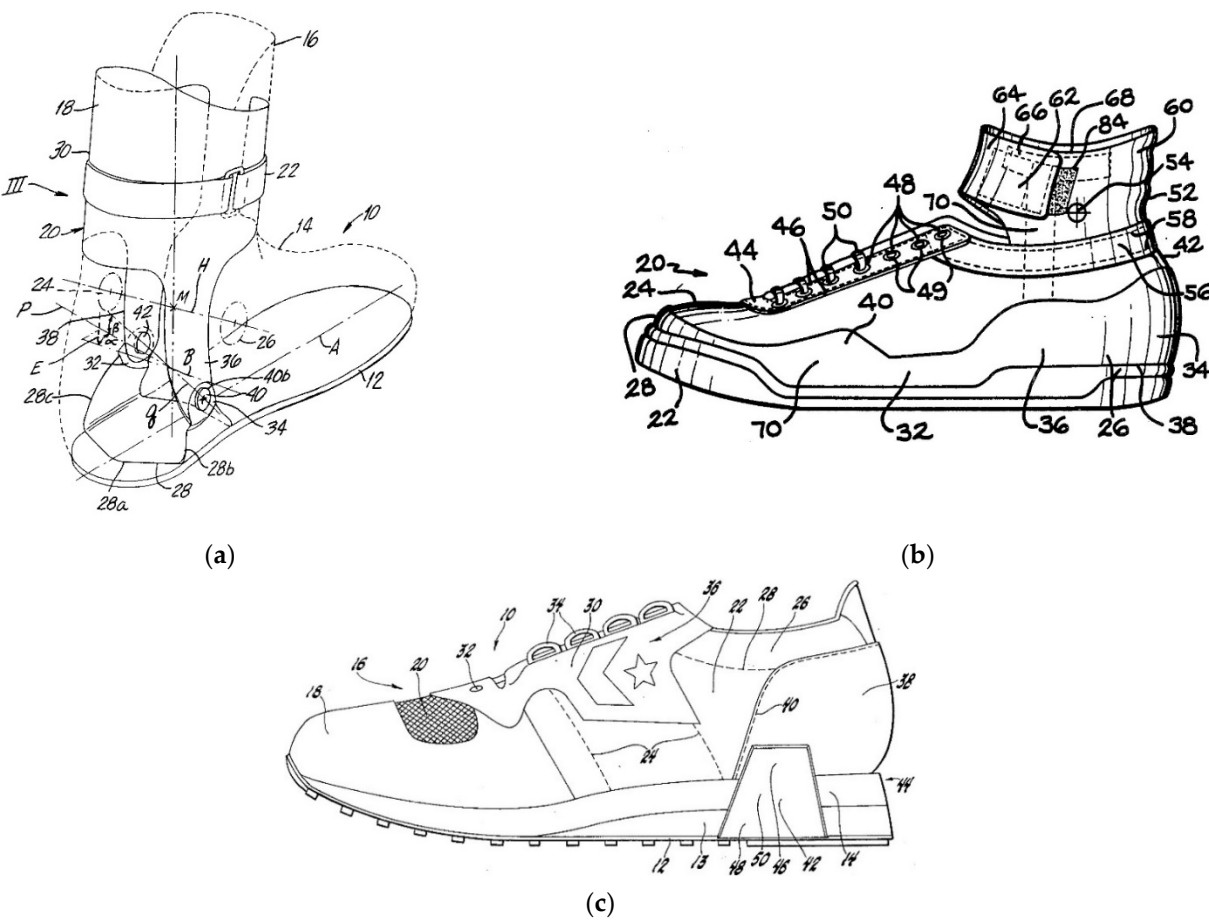

(**a**)　　　　　　　　　　　　　　　　　　　　　　　　(**b**)

(**c**)

**Figure 13.** Patents of Orthosis for sports use: (**a**) Shoe, in particular, Sport Shoe or Orthopaedic Stocking with Ankle Stabilisation [63], (**b**) Athletic Shoe [64], and (**c**) Running Shoe Sole with Heel Tabs [65].

*5.4. Selected Patents on Others*

Of the 150 granted patents analysed, 3 were categorised as others. One patent of interest in this category is the smart device invented by K-M. Chan, T.P.D. Fong, and S.H.P. Yung [66], published in 2010. It describes a smart wearable device for preventing ankle sprain injury (as shown in Figure 14). It comprises three parts: (1) a sensing part configured to sense ankle inversion velocity; (2) a data analysing component configured to

determine if the motion detected was a sprain motion and generate a trigger signal if the data received exceeds a threshold indicative of sprain motion; and (3) a stimulating part configured to stimulate the lower limb muscles in response to the trigger signal transmitted from the analysing part.

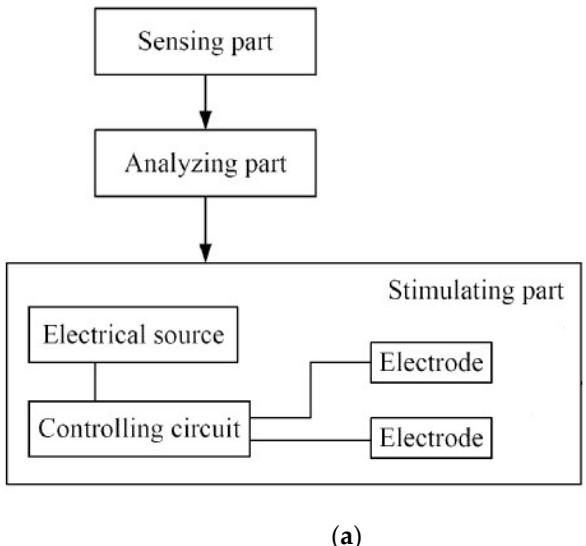

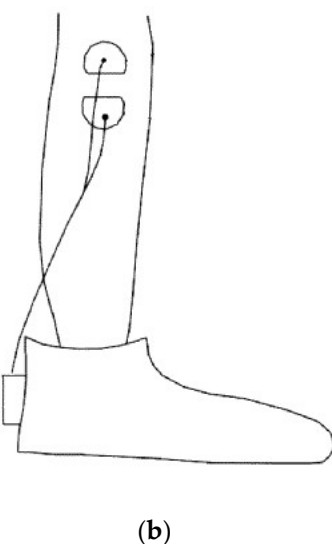

(**a**)                                                                 (**b**)

**Figure 14.** Methods and Devices for preventing ankle sprain injuries [66]: (**a**) schematic block diagram of the device and (**b**) device placement on lower limb, according to an embodiment of the present invention.

## 6. Comparisons with Existing Literature

The effectiveness of taping as a method of preventing ankle sprain has been widely suggested in the academic literature [38,67–74]. Willeford et al. [75] also suggested that taping and bracing provide equal range of motion restriction. However, there has been a significantly small number of patents on taping compared to ankle bracing. A possible reason for this is that patents of medical or sports tape are usually for general use and not patented specifically for the prevention of ankle sprain. In addition, methods of ankle taping do not fall under patentable subject matter.

There were no patents on prevention training or devices for prevention training. Devices used for prevention training are normally for general use and not specifically for ankle sprain prevention, hence the absence of patents specific for the purpose. Smartphone apps to support ankle sprain prevention training also do not fall under patentable subject matter.

There have also been suggestions in the literature on the use of ankle braces as an emerging prevention method [6,76]. However, the number of granted patents on braces, as seen in Figure 8a, has not seen any noticeable increase, which suggests the use of ankle braces is still in the early technological readiness level and has not gained traction among industries and practitioners. There is also a significantly large number of patents on orthosis (as seen in Figure 7). Despite the difference in opinions on the effectiveness of orthosis among researchers, there is a strong appetite for protecting technological development on orthosis, indicating a strong confidence among some quarters in orthosis for preventing ankle sprain.

Additionally, the emergence of devices that can 'sense' an ankle sprain just before it occurs can been seen in our patent review. This combines traditional prevention methods and sensor technology. The Japanese patent JP2015150437 [60] on a brace and the US Patent US20100042182 [66] both include components that could detect if the user is about to sprain their ankle. Previous academic systematic reviews have yet to include this as an emerging method. However, there have been studies [77–83] within the last 20 years on the development of devices and methods that can sense ankle sprain.

### 7. Conclusions

Prevention of ankle sprains is important, given the frequency of occurrence of an ankle sprain, the risk of reinjury, and its long-term effects. A patent landscape review has been successfully conducted in this review to give an overview of the area, as well as to study the technological development of ankle sprain prevention methods by looking into patent filings worldwide. In the past decade, China and the United States of America have shown great interest in developing ankle sprain prevention methods. Of the 426 patents that were selected for review, 43% of the published patents are from China, while 29% are from the United States of America, with China showing an upsurge of new patent filings from 2009. The types of ankle sprain prevention methods have been classified into braces, orthosis, tape, and others. Approximately 51%, or 76, of the 150 granted patents are associated with braces; 44%, or 66, are related to orthosis; 3%, or 5, are related to tape; and the remaining 2%, or 3, are associated with others, showing that innovators are leaning towards braces and orthosis as the main prevention methods for ankle sprains, with more emphasis on orthosis in recent years. There is also a growing interest in orthosis for sports use, particularly for general athletics, American football, and baseball. This is contrary to the emphasis on braces and tape and the reduced emphasis on orthosis in systematic reviews of the academic literature. Most current designs of braces and orthosis focus on allowing free mobility of the foot, while effectively shielding the ankle and reducing the risk of ankle sprains. However, there is a lack of variety among the designs and methods described in the patents on tape, despite the efficiency of taping in preventing ankle sprains, according to academic literature. Additionally, the publication and granting of a patent on smart wearable devices that can sense ankle sprains implies early commercial interest in smart wearable devices for ankle sprain prevention.

However, this study has several limitations. Firstly, the patent documents obtained for this review are limited to the patents available on the PATENTSCOPE database. WIPO's database provides only patent applications that have been filed under the PCT system and those patents from participating national and regional patent offices. In addition, new national patent collections and Global Dossier information that have been made available to the PATENTSCOPE database after 5 September 2020 would not have been included in this patent review. Secondly, our search was limited to patent documents that claimed to specifically prevent ankle sprain. Devices that would be of general use would not have been included in the search, leaving out many patents of devices that could be used for prevention of ankle sprains. Lastly, there is a possibility of human error, as the patent documents were filtered manually to remove patents of the same patent family.

Studies on the commercially available ankle sprain prevention devices may present interesting future directions and extensions to the current work. The commercially available ankle sprain prevention devices may be matched to see if they emanate from either the available patent documents or academic literature.

**Author Contributions:** Conceptualisation, H.S. and P.E.A.; methodology, N.F. and A.O.; validation, N.F. and A.O.; formal analysis, N.F. and AO.; investigation, N.F. and A.O.; writing—original draft preparation, N.F. and A.O.; writing—review and editing, H.S. and P.E.A.; visualisation, N.F. and A.O.; supervision, H.S. and P.E.A.; funding acquisition, H.S. All authors have read and agreed to the published version of the manuscript.

**Funding:** This research and the APC was funded by Universiti Brunei Darussalam Research Grant No: UBD/RSCH/1.3/FICBF(b)/2020/012.

**Data Availability Statement:** Publicly available datasets were analysed in this study. This data can be found here: www.patentscope.com (accessed on 5 September 2020).

**Conflicts of Interest:** The authors declare no conflict of interest.

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
