# Peer review of "Patent Landscape Review on Ankle Sprain Prevention Method: Technology Updates"

_inventions, doi:10.3390/inventions8020053_

Round 1

Reviewer 1 Report (New Reviewer)

This paper, in the form of a patent landscape review, describes what has been patented in the area of ankle sprain prevention. In the introduction and background sections, the literature review as well as the statistics have been very well documented. The paper is organized in a logical and clear flow, and the methodology is well designed and described, as well as the results are well-presented. However, my first concern is related to patent families (interrelated patent documents - with the same technical detail - that were published at different times or in different countries). The second concern is related to the use of only the PATENTSCOPE database as a resource. Meanwhile, this WIPO's database provides only international patent applications filed under the PCT system and also provides access to a few national and regional collections. While other commercial databases use multicountry research services and have access to all patent documents at the international, regional, and national levels through databases provided by WIPO, all regional offices, and all national offices, respectively. Further, more would be needed to make this a valuable contribution to the literature. I suggest that the authors revise this paper.

Comments are outlined below:

1.        How much of this type of work (in this area) is completed prior to your work?

2.       The authors should distinguish between the abstract section and the conclusion section (e.g., the last sentences: Lines614-617 and Lines20-22.

3.       At the end of the introduction, the authors should present a summary of the methodology followed in this paper.

4.       Based on the earliest priority date concerning patent documents in relation to ankle sprain prevention methods, the authors should describe the first patent application or granted patent. The aim is to demonstrate the start of patenting in this area.

5.       Figure 1 should include patent families (simple and extended) to delineate interrelated patent documents (for the same invention) that were published at different times or in different countries.

6.       Section 5 could have been even more useful if any of the patented technologies had been used in actual biomedical devices. This section could have included examples of commercial applications. If the authors can add such information, this manuscript will be really useful.

7.       The authors should adopt "patent documents" (granted patents, patent applications, limited patents, patents of addition, amended patents, etc.) as a generic term for all patent types (full patents). But generally, we use only the following types: granted patents and patent applications. In the manuscript, the authors should replace "published but not yet granted" by "patent applications" and "patents have been granted" by "granted patents". Examples: lines 227-229 ; Figure 1 ; Figure 2 ; lines 269-270, etc.

8.       The authors should be careful with different dates for patent documents. There is a difference between the filed date, the publication date, and the granted date. The authors should adopt a publication date in the manuscript, and if it’s necessary, they can include the filed date or granted date in brackets.

9.       The authors should avoid the patent number in the manuscript. This should be replaced by "inventor name" or "inventor name et al.", publication year…

10.    Example: line 468:  US patent US5288286 [52], granted in the year 1994, describes an adjustable pressure cast for orthopaedic injuries…” should be written as:  in 1994, Davis invented (or proposed) an pressure cast for orthopaedic injuries…”.

11.     The list of references should contain information about the patent document (if granted patent or patent application, e.g., [52] A. D. Davis, "Adjustable Pressure Cast for Orthopedic Injuries," Granted Patent US5288286A, 1994. Also, the patent number should be like US5288286A, JP2015150437A, JP6664877B2, etc.

12.    Line 490: JP2015150437 is a patent application, not a granted patent. The Japanese granted patent is the patent family JP6664877B2. Please check the manuscript to see if all cited patent documents are patent applications or granted patents.

13.    For discussion: How has this field evolved? What are the distinct trends? Can anything be inferred about the future?

14.    Finally, the conclusions section needs comments on the limitations and perspective of this study.

15.    I found this recent reference interesting for the patent landscape review. The authors should use it to improve their paper: Exploring the patent landscape and innovation of hydrogel-based bioinks used for 3D bioprinting. Recent Advances in Drug Delivery and Formulation 2022, 16, 145-163, doi:10.2174/2667387816666220429095834. Also, the authors should cite other recent references in relation to “patent analysis” and “patent landscape”.

Author Response

Reviewer 2 Report (New Reviewer)

This study provided a patent review on the technology of wearable ankle sprain prevention, which showed an interesting topic and some new insights for the industrial decision-makers.

Besides, in my opinion the paper could be improved in the following aspects.

(1) The paper can draw a figure of research framework that indicates the main components of patent landscape analysis.

(2) The authors need to report the time when the study finishes the patent retrieval in methodology, since the data may change as time goes by.

(3) Where is the citation [101] in line 344? The literature seems not in the reference list.

(4) What enlightenments can be formed in the research conclusions? For example, technology research and development, patent application, etc.

Round 2

Reviewer 1 Report (New Reviewer)

Accept in present form.

This manuscript is a resubmission of an earlier submission. The following is a list of the peer review reports and author responses from that submission.

Round 1

Reviewer 1 Report

Abstract:

1.     Line 19: Add sprain after the word ankle.

Introduction:

1.     Line 27-28: This statement is highly debatable. Often, low back pain, arthritis, and trauma are cited as the most frequent musculoskeletal injuries. Consider softening the language here.

Background

1.     Much of this section should be removed. Unless the information contained herein benefits the reader and our understanding of the technology you located in your patent search, much of this is superfluous. This includes removing Lines 58-63, potentially all of the anatomy section (line 68-89), lines 102-118, and lines 127-134.

2.     Table 2: There should be a differentiating line between grade 1, 2, and 3 signs and symptoms. Otherwise it looks like all of this goes together.

3.     Again, the discussion about CAI seems a bit irrelevant. Does this inform the patent search and the technology you found? It seems like the answer is no.

4.     The current methods of preventing ankle sprain section should likely just be combined with your results. This is in essence what you are searching for. Seems out of place as written. Not clear why there is a long introduction to the current technology when the point of the paper is finding this via patent searches.

Discussion

1.     It’s not entirely clear what the takeaway message is from your search. This should be more thorough and clearer with respect to wear the technology is and where it is heading.

Reviewer 2 Report

 First of all, to thank the effort and dedication in contributing to the updating of the scientific field through the generation of publications.

In my opinion, it is not a topic of great interest to the scientific population, since there are many publications that deal with the same topic. In addition, in my opinion, it is a very extensive work to contemplate a paper. Perhaps it is better to make a manual or something similar. It must be taken into account that this is not an analytical and experimental study, so it does not add new data to what is already published in the scientific literature. Taking into account that it is a systematic review, it has very old bibliographic citations (1994, 2008, etc). I think that in order to be published as a paper, it will be necessary to redo the work, update it and shorten it. For all these reasons, I think it is not suitable for publication.